# Limits to reproduction and seed size-number trade-offs that shape forest dominance and future recovery

The relationships that control seed production in trees are fundamental to understanding the evolution of forest species and their capacity to recover from increasing losses to drought, fire, and harvest. A synthesis of fecundity data from 714 species worldwide allowed us to examine hypotheses that are central to quantifying reproduction, a foundation for assessing fitness in forest trees. Four major findings emerged. First, seed production is not constrained by a strict trade-off between seed size and numbers. Instead, seed numbers vary over ten orders of magnitude, with species that invest in large seeds producing more seeds than expected from the 1:1 trade-off. Second, gymnosperms have lower seed production than angiosperms, potentially due to their extra investments in protective woody cones. Third, nutrient-demanding species, indicated by high foliar phosphorus concentrations, have low seed production. Finally, sensitivity of individual species to soil fertility varies widely, limiting the response of community seed production to fertility gradients. In combination, these findings can inform models of forest response that need to incorporate reproductive potential.

The emergence of extreme disturbance as a global change phenomenon[1,2] highlights the need to understand how tree fecundity influences forest regeneration. Drought-induced forest diebacks are now observed in regions where they were rare in the past[3]. Forest stands shaped by non-destructive surface fires are experiencing catastrophic crown fires, with post-burn seed production limited to survivors in unburned fragments (Fig. 1a). Species capable of vegetative regrowth will contribute to reforestation[4], but colonization and seedling success in many landscapes will depend on numbers and sizes of seeds from surviving trees. Large seeds that are well-provisioned for seedling survival are expensive to produce, apparent to seed predators, and dependent on animals for dispersal. If the costs of producing larger seeds are offset by producing fewer of them[5,6], then reproduction could be capped by a size-numbers trade-off. If this cap depends on resources, then landscape fertility gradients and differing nutrient requirements of species play important roles. While species differences in seed size are readily observed and the subject of a large literature[7–9], seed numbers are not. At least since Darwin pondered the rapid ascendance of angiosperms in the Cretaceous[10,11], seed production has been recognized as a fundamental component of fitness[12–15] that lacks systematic quantification for trees, the planet's dominant life form[16–18]. Here, we provide a global quantification of tree reproduction from more than 12.1 million tree-years of observations on 714 species. Results show that species differences are not bound by a strict size-numbers trade-off, instead ranging over ten orders of magnitude and mediated by foliar nutrients and soil fertility, with phylogeny accounting for some of the broad variation in fecundity.

A size-numbers trade-off could limit variation between species through allocation constraints associated with producing big seeds versus many of them[5,19,20]. Because seed production depends on tree size[21], we standardize seed production per m² of tree basal area. We define individual standardized production (ISP) as (mass per seed) × (mean seeds per tree basal area) or

$$\text{ISP}_{is} = m_s \times F_{is} \quad (1)$$

for tree $i$ of species $s$ with standardized seed production $F_{is} = f_{is}/b_i$ for a tree with basal area $b_i$. The estimate of seed number $f_{is}$ incorporates year-to-year variation in seed production. Species seed production (SSP) is the expectation of ISP over trees of a given species ($\text{SSP}_s = E_i(\text{ISP}_{is}) = m_s \times E_i(F_{is})$, Methods). If the benefits of large seeds for attracting mutualist dispersers and increased seedling survival[22] is offset by the cost of producing fewer of them, then SSP does not vary with seed size. But it does vary if producing larger seeds increases or decreases the net reproductive output summarized as ISP. This departure from the 1:1 trade-off is expressed on the proportionate ($\log_{10}$) scale as

$$\text{SSP}_s \propto m_s^{\beta}$$
$$\log_{10}\text{SSP}_s = \alpha + \beta\log_{10} m_s \quad (2)$$
$$\log_{10}\hat{F}_s = \alpha + (\beta - 1)\log_{10} m_s$$

where $\hat{F}_s = E_i(F_{is})$ is the expected seeds per basal area for species $s$, $\alpha$ is the $\log_{10}F$ for a species having seed size equal to 1 g, and $\beta$ is the proportionate change in $\hat{F}_s$ for a proportionate increase in seed size. [Subscripts are hereafter omitted to reduce clutter.] The size-numbers trade-off predicts that $\beta = 0$ or, equivalently, a slope of $-1$ for $\log_{10}m_s$ in Eq. (2). A value of $\beta > 0$ means that numbers do not decline with seed size as steeply as predicted by the trade-off, i.e., a net (proportionate) gain in reproductive effort of $\beta - 1$. Values of $(\alpha, \beta)$ are unknown, as attempts to quantify species differences in SSP have been limited geographically and to few species[20,23] due to missing information on numbers of seeds produced. Because large seeds may confer a competition benefit to seedlings within a crowded understory, while small seeds that are wind-dispersed can promote colonization of distant sites[14,24], we further evaluated the potential for a competition-colonization trade-off, examining the hypothesized link between competitive ability and wood density[25,26].

Species differences in seed size versus numbers could be affected by the costs of auxiliary reproductive structures[27]. Total reproductive effort includes seeds and flowers supplied with nectar and nutrient-rich fleshy fruits that attract dispersers (e.g., Rosales, Ericales, Cornales, Aquifoliales, Magnoliids, and Laminales). It includes cones defended with wood, armaments, and prodigious resin flows (e.g., Pinales) (Fig. 1b). Comprehensive mass and nutrient data for cones, fleshy fruits, nuts, capsules, and samaras are absent from trait data bases like TRY[28], nor is there a transparent means for comparing reproductive costs across diverse structures and provisioning. For these reasons, species differences in auxiliary structures like fruits and cones are compared here on a qualitative basis.

Limits to SSP could depend not only on the cost of auxiliary structures associated with seed production, but also on resources. Studies of nutrient effects are limited geographically and taxonomically, and results are equivocal[29–33]. Through the assumption that reproductive allocation scales with net primary production (NPP)[34–37], current models assume that resources stimulate fecundity, without consideration of responses that

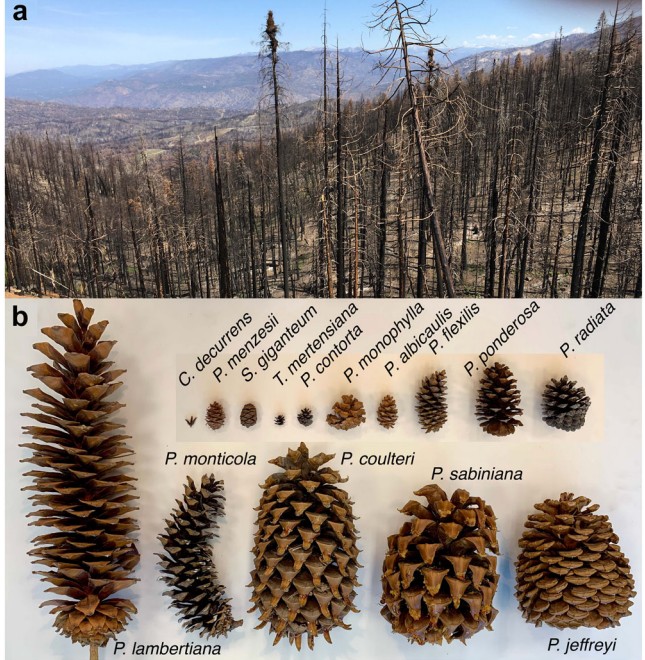

**Fig. 1 Seed production quantifies forest regeneration potential.**
Regeneration of forests devastated by multi-year drought and fire depend on a vastly diminished seed supply. **a** Seed production is limited to unburned landscape fragments in the Sierra Nevada mixed conifer zone following 2020 burns at a Masting Inference and Forecasting network (MASTIF) and National Ecological Observatory Network (NEON) site (Shaver Lake, CA). **b** Total reproduction includes not only seeds, but also defenses, including wood, spines, and resin flow in conifer cones; examples from the heavily burned Sierra Nevada and Coast ranges include *Calocedrus decurrens, Pinus albicaulis, P. contorta, P. coulteri, P. flexilis, P. lambertiana, P. monophylla, P. monticola, P. ponderosa, P. radiata, P. sabiniana, Pseudotsuga menzesii, Sequoiadendron giganteum,* and *Tsuga mertensiana*. Mass fractions for seeds to seeds plus cones ranges from 3% for *P. radiata, P. contorta, P. coulteri,* and *P. sabiniana* to 61% for *C. decurrens*. The largest cone in **b** (*Pinus lambertiana*) is 46 cm. Photo credits: James S. Clark and Jordan Luongo.

might come from variation within species (here, ISP) versus between species (here, SSP). Nutrient additions in orchard practice use ratios intended to balance demands for nitrogen (N), phosphorus (P), and potassium (K)[29,38], the latter being especially important for fruit yield in agriculture, but less frequently reported in ecological studies[39]. Agricultural experience makes it clear that high-N fertilization can *decrease* reproductive yield[40,41] due to allocation to vegetative growth. Variation between species could come from their differing nutrient demands, wherein species with high average foliar nutrients tend to occupy fertile sites[42,43] and, thus, might be expected to produce more seed. Alternatively, species with high nutrient demands might preferentially allocate to growth and defense.

Variation within species could come from phenotypic plasticity as genotype × environment ($G \times E$) interactions. Species bearing large seeds and nutrient-rich fleshy fruits might be especially responsive to fertility variation, but data are lacking. Observed positive correlations between foliar nutrients and seed production in the same individuals over time[44,45] does not mean that there is a positive association in nutrients and seed production taken over trees growing at different fertility levels. Our use of ISP (standardized for tree size) controls for responses that come from the fact that fertilized trees grow faster and thus are larger[21,46], thus helping to isolate fertilization responses[47] that are caused by increased seed production for a given tree size. Taken together, reproductive responses to fertility might be positive or negative, depending on allocation to growth and reproduction.

As mentioned above in the context of current model assumptions, the implications of fecundity for landscape recruitment has to combine ISP with the turnover of species across environmental gradients, here termed community seed production (CSP) for the amount of seed produced per area of forest or woodland. Trees of the same species might produce more or less seeds on the richest sites, depending on partitioning between reproduction and growth[15]. Species turnover influences CSP if the resource-conservative species that dominate nutrient-poor sites[42,43,48] differ in their ISP from those on fertile sites. To be clear, we do not obtain CSP as an extrapolation of counts in seed traps. That practice would generate huge error due to the precise locations of seed traps, especially their siting directly below an individual that produces large seeds that almost never enter seed traps were they not directly below the tree. Reliable estimates of seed production per ha requires summing the production over all trees in the plot divided by the area of the plot (see Methods); this is the method used here for CSP.

This synthesis addresses four basic questions, framed as hypotheses: (A) a size-numbers trade-off constrains species differences in SSP; (B) high auxiliary costs related to fleshy fruits and cones (Fig. 1b) and high nutrient requirements result in high or low SSP, depending on allocation trade-offs involving vegetative growth and defense; (C) ISP increases with soil fertility or not, again depending on species-specific resource allocation trade-offs; and (D) CSP increases or decreases with soil fertility due to the combined effects of within-species sensitivity (hypothesis C) and species turnover across gradients. Confirmation or rejection of these hypotheses is central to understanding the evolutionary selection pressures that have shaped the dominant forest trees of the world and is likely key to mitigating their future losses and recovery with global change.

To test these hypotheses, we synthesize tree fecundity, foliar nutrients, and cation exchange capacity (CEC) (Supplementary Fig. 1)[49], a widely used fertility indicator related to soil capacity to bind exchangeable ions (e.g., calcium, magnesium, sodium, potassium, ammonium, and iron) that are essential to plant function[50,51]. CEC has an advantage over soil carbon or nitrogen that are high not only on fertile sites, but also in saturated,

nutrient-poor peatlands that dominate many important ecosystems[52]. The Masting Inference and Forecasting (MASTIF) network provides a global synthesis of individual tree-year observations and modeling framework that estimates effects on fecundity that accounts for dependence in data. Low signal-to-noise in seed production data requires massive sample sizes to estimate effects[21], represented here by 12.1 M observations on 714 species from all vegetated continents. Valid estimates of fecundity responses to foliar nutrients and CEC accommodate dependence in seed observations between trees and seed traps and within trees and seed traps over time, the quasi-synchronous, quasi-periodic variation known as *masting*. Individual tree-year observations are modeled jointly in a hierarchical Bayesian state-space model detailed in Clark et al.[53].

## Results

Rather than being constrained to constant values by a size-numbers trade-off (hypothesis A), species seed production per basal area (SSP) spans ten orders of magnitude and increases with seed size ($\beta = 0.546 \pm 0.042$, Fig. 2a). Across all species in the synthesis, the expected seed number per m² of basal area for a seed size of 1 g is $E(10^\alpha) = 19,700 \pm 2920$ (Eq. (2) and Methods section). For a small-statured species this amounts to 0.62 kg at a diameter of 20 cm. For a large species, this is 15 kg for a diameter of 100 cm. For a small-seeded species of, say, 0.001 g (e.g., *Betula*), this amounts to annual averages of > 50,000 seeds at 40 cm and > 300,000 seeds at 100 cm. If SSP is regulated by a proportionate sacrifice in seed size as the cost of producing more of them, then SSP would not vary with seed size (dashed line in Fig. 2a). Equivalently, there would be a proportionate decline in seed numbers with seed size (slope $= -1$, dashed line in Supplementary Fig. 2). Instead, seed numbers decline with seed size at less than half the proportionate rate for a slope of $\beta - 1 = -0.454 \pm 0.042$ (Supplementary Fig. 2). This departure from expectation comes from variation in SSP that is related to species traits and partly captured by phylogeny.

Species differences in SSP include a phylogenetic component (Fig. 2b, *Pagel's* $\lambda = 0.60$, $p < 10^{-9}$, $n = 482$). Group seed production (GSP, methods) was used to compare fecundity between different taxonomic groups, with higher average values for angiosperms than gymnosperms (Supplementary Fig. 3a). The possibility that auxiliary costs weigh against SSP (hypothesis B) is consistent with low SSP for cone-bearing Pinales (brown labels in Fig. 2b). In angiosperms, the Fagales have uniformly high SSP, despite seeds ranging from small *Betula* ($1-6 \times 10^{-4}$ g) to large *Quercus* (0.7–7 g). *Cecropia*, *Miconia* and the Fabales have notably low SSP. Angiosperms with fleshy fruits (green labels in Fig. 2b) do not exhibit lower SSP than other angiosperms on the whole (Supplementary Figure 3b). Some groups are predominantly high (e.g., Ericales/Cornales), low (e.g., *Vismia*, *Cordia*, and Campanulids) or mixed (e.g., Rosales and Magnoliids).

At the species level, SSP declines with increasing mean foliar P after controlling for phylogeny and leaf habits (hypothesis B). The surface in Fig. 3 decreases four orders of magnitude from > 10,000 at the lowest foliar $P$ levels to < 10 g per m² basal area at the highest foliar P. By contrast, foliar N has no discernible effects on SSP. Broadleaf species span the widest range of concentrations and contribute most to the declines in SSP with increasing P. Wood density did not explain SSP variation (Supplementary Table 5).

Within species (hypothesis C), phenotypic plasticity in ISP to soil fertility, hereafter $\beta_{cec}$, varies widely between species, but with phylogenetic coherence (Fig. 4). In the well-represented Fagales, fertility responses in *Castanea* and the white and cerris oaks (sections *Quercus* and *Cerris*) are negative, while red oaks (section

**b**

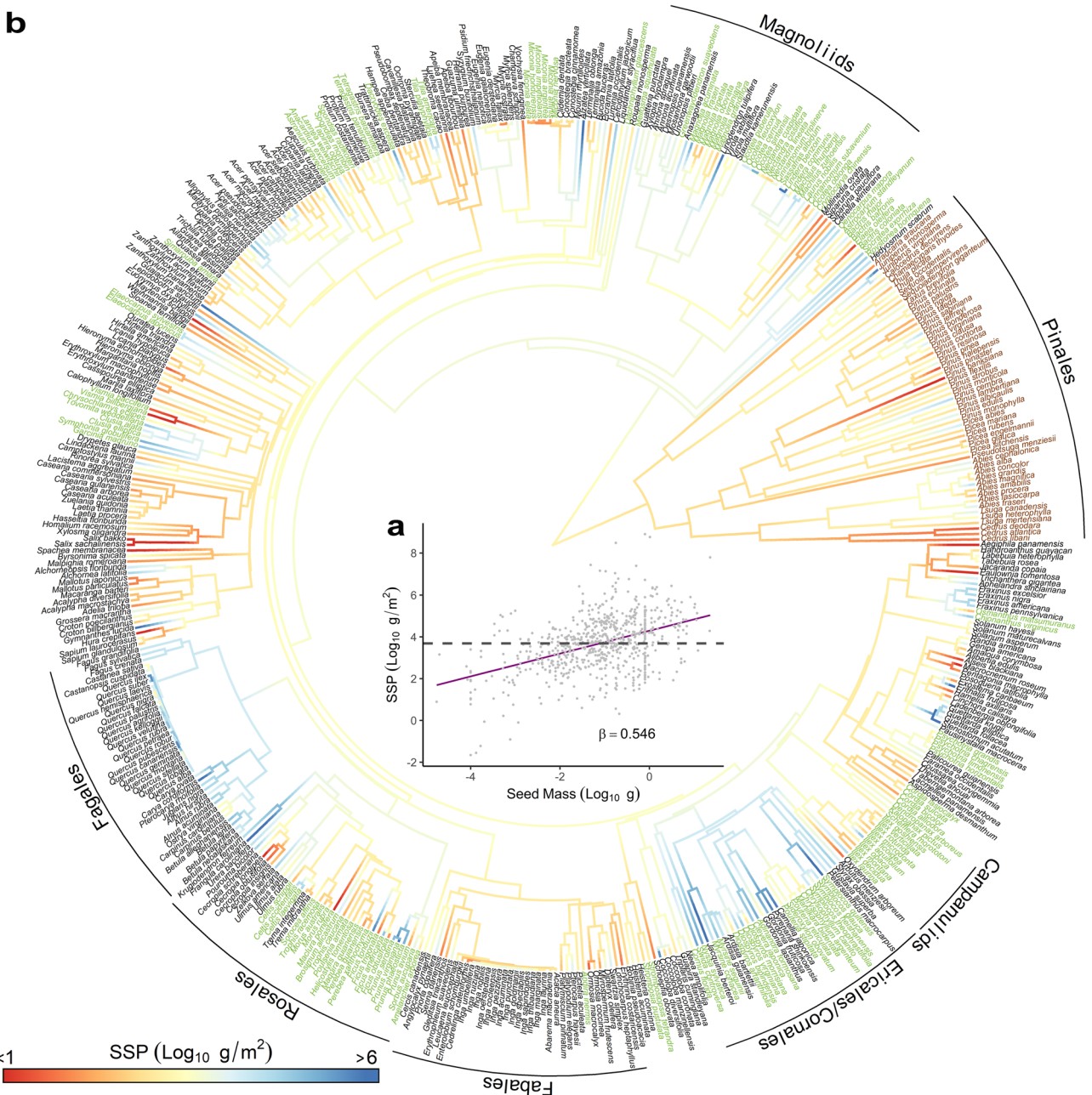

**Fig. 2 seed size-number trade-offs and species seed production. a** Species seed production (SSP, g seed per $m^2$ tree basal area) is not constrained by the strict size-number trade-off (dashed line with a slope of zero). Instead, it varies over ten orders of magnitude and has a positive correlation with seed mass across 714 tree species ($\log_{10}SSP = 4.29 + 0.546 \cdot \log_{10}m$, $R^2 = 0.189$, $p < 10^{-15}$, $n = 714$). **b** SSP exhibits phylogenetic coherence for 482 species having phylogeny data (68% of species). Brown and green text highlight species that produce coniferous cones and fleshy fruits, respectively. The phylogenetic signal is estimated using *Pagel*'s $\lambda = 0.60$ ($p < 10^{-9}$, $n = 482$).

*Lobatae*) are positive. This divergence coincides with the tendency for many red oaks (*Lobatae*) to occupy more fertile soils[54]. Increasing CEC reduces ISP in the angiosperm groups Fabales, Magnoliids, *Acer*, and *Fraxinus*, while stimulating ISP in Ericales/ Cornales, and having mixed effects in Rosales and Betulaceae. Gymnosperm responses are primarily decreasing for *Tsuga*, *Picea*, *Cedrus*, and soft pines (section *Haploxylon*), and mixed for hard pines (section *Diploxylon*) and *Abies*. Despite species variation, the reconstructed ancestral lineage for gymnosperms is negative (purple ancestral branches marked by red dashed line in Fig. 4). By contrast, there is no angiosperm-wide tendency to produce fewer seeds on high-fertility sites. Foliar nutrients do not explain these species differences in response to fertility gradients.

Community seed productivity (CSP) does not respond to soil fertility (hypothesis D), due to the neutralizing effects of diverging responses within species that occupy a range of CEC levels (Supplementary Fig. 4) and to the turnover of species across gradients. The differing sensitivity to CEC within species combined with the species turnover results in seed production that is as high on low-fertility as on high-fertility sites.

**Discussion**

Species seed production (SSP), measured as seed size × seed number per basal area, varies over ten orders of magnitude (Fig. 2), far from the equity expected from a strict size-numbers trade-off. Ecologists have long recognized a potential trade-off

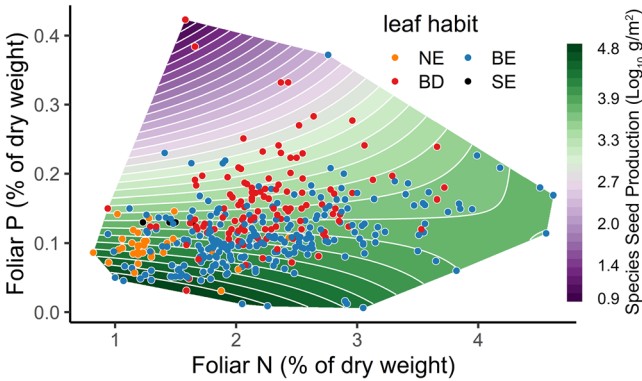

**Fig. 3 Effects of foliar nutrients on seed production.** Effects of foliar nitrogen (N) and phosphorus (P) on SSP (g seed per m² tree basal area) from the model in Supplementary Table 1 plotted for the broadleaf deciduous leaf habit (other leaf types exhibit same patterns). The convex hull for the surface is restricted to the data coverage. Symbols indicate leaf habit, including broadleaf deciduous (BD), broadleaf evergreen (BE), needleleaf evergreen (NE), and scalelike evergreen (SE).

between producing few large seeds, each provisioned to attract dispersers and/or promote high seedling survival, versus many small seeds that might offset low survival with expanded opportunities to reach suitable sites[5,6,20]. Evaluating whether or not this compromise between size and numbers limits species variation in SSP is made possible by a large representation of 12.1 M tree-years from 714 species and an analytical framework that admits tree attributes (size and competition), environmental variation, and dependence between trees and within a tree across years[53]. Results show that the increased seed-mass production realized by species that invest in large seeds is mediated by resource access and the auxiliary costs of woody cones.

The expected annual productivity of 19,700 ± 2920 seeds per m² basal area at a seed size of 1 g provides a benchmark for basic demographic and life-history understanding that has thus far been absent for trees. For a moderate size tree (e.g., 30 cm diameter) that produces seeds of a moderate size (e.g., $10^{-2}$ g) this amounts to annual mean production of > 10,000 seeds. The magnitude of this estimate puts this life form far outside the range of most species, including most herbaceous plants[8]. This massive output certainly contributes to a foundational role of masting in many forest food webs[55]. It is expected to drive inordinate selection pressures required to offset seed input with mortality rates that would be needed for quasi-stable population densities over time.

There is a size-numbers trade-off, but it is not symmetric: species that invest in large seeds produce more seeds than expected from a strict 1:1 trade-off. The shallow slope between seed number and size (Supplementary Fig. 2), roughly half that expected for a strict trade-off[56,57], suggests a cost savings as reproductive investment is packaged into a smaller number of larger seeds. If the cost of reproduction increases as $\beta = 0.55$, or roughly the square root of seed mass (Eq. (2)), then the benefit for large seeds increases according to the slope in Fig. 2a. A fixed cost associated with the production of each seed would be consistent with this shallow slope; the diversity of structures used to support reproduction suggests multiple contributing factors.

Of course, the slope of the size-numbers relationship *between species* does not represent strength of selection for size and numbers *within species*, which depends on a broad range of traits summarized, in part, by phylogeny. The advantages of producing larger/fewer acorns in *Quercus* cannot be equated with the advantages of producing larger/fewer fleshy fruits or cones, due to the differing costs and benefits associated with each of these structures. Low SSP in gymnosperms is consistent with the

investment in cones that can exceed seed mass by 30-fold in Fig. 1b. But the generally low SSP in gymnosperms does not imply a fitness disadvantage for large cones. Well-defended seeds certainly arose under selection pressure for the advantages they confer against predation[58]. The divergent evolutionary options available to angiosperms and gymnosperms may have constrained reproductive potential for these two groups in different ways; there is no evidence that angiosperms that produce expensive, fleshy-fruits realize lower SSP than other angiosperms (Fig. 2b and Supplementary Fig. 3b).

Phylogenetic differences (Fig. 2b and Supplementary Fig. 3) bear on Darwin's 'abominable mystery'[10], the advantages that might explain the explosion of angiosperms in the Cretaceous. The rise of angiosperms has been attributed to rates of growth[59], photosynthesis[60], and diversification[61], all of which tend to be higher than in gymnosperms. Capacities to harbor nutrients and generate positive feed-backs with soil fertility through litter production[62] and to reproduce vegetatively[63] could contribute to angiosperm success. Aridification at the end of the Eocene may have contributed to gymnosperm extinctions[11]. While all of these differences are plausible contributions to angiosperm success[64], the fact that angiosperms dominate gymnosperms in SSP represents a more direct connection to fitness than most of these mechanisms.

The negative effect of fertility on seed production emerges in the low SSP for P-demanding species (Fig. 3). Global change studies have steadily improved our understanding of fertility effects on tree growth, foliar nutrients, and primary productivity[65–67], while tree fecundity has remained inconclusive[32,33]. This synthesis clarifies three elements of the fertility-fecundity relationship. The species-level association between high foliar P and low SSP (Fig. 3) is consistent with a species-level trade-off between growth and reproduction[15,16]. P-rich ribosomal RNA required for rapid cell division[68,69] can stimulate growth at the expense of reproduction. Moreover, seed production is expensive in resource-acquisitive species[23] if nutrient concentrations in seeds increase with foliar nutrients[70]. The decline in SSP with foliar P is consistent with horticultural evidence that fertilization can increase foliar nutrients and stimulate vegetative growth at the expense of both quantity and quality of crop yield[41,71,72]. However, a plant's response to nutrient addition[45,73] need not agree with species differences in nutrient demand. Potassium (K) promotes reproductive yield in crops[74,75], but data for trees are limited. Second, within-species responses to CEC (i.e., $\beta_{cec}$) are phylogenetically conserved despite variation in magnitude and sign across species. The negative effect of CEC on seed production in gymnosperms (purple shading that dominates the ancestry of this group in Fig. 4) is consistent with conifer dominance in infertile environments[62,76] and their resistance to angiosperm expansion on poor soils[77,78]. Finally, the fertility gradients that contribute to trends in net primary productivity (NPP) do not translate to trends in fecundity due to the combined effects of within-species responses (Fig. 4) and the turnover of species across these gradients (Supplementary Fig. 4).

Seed size versus numbers is among the most fundamental constraints on population persistence in competitive stands[79–81], on landscapes subject to large-scale disturbance[82,83], and where colonization requires long-distance dispersal[84,85]. Just as reproductive constraints may have contributed to global transformations in earth history, they will undoubtedly shape the forests that emerge from expanding diebacks, disturbance, and human exploitation (Fig. 1a). Species and forest types are threatened by regeneration failure following climate change, timber harvesting, and other disturbances. Globally, less than 7% of forests are planted by humans, but, over the last decade, naturally regenerated forests have decreased by 8 million ha per year[86]. The expanding scale of forest losses might increase reliance on artificial regeneration[86,87], with attendant costs to forest diversity.

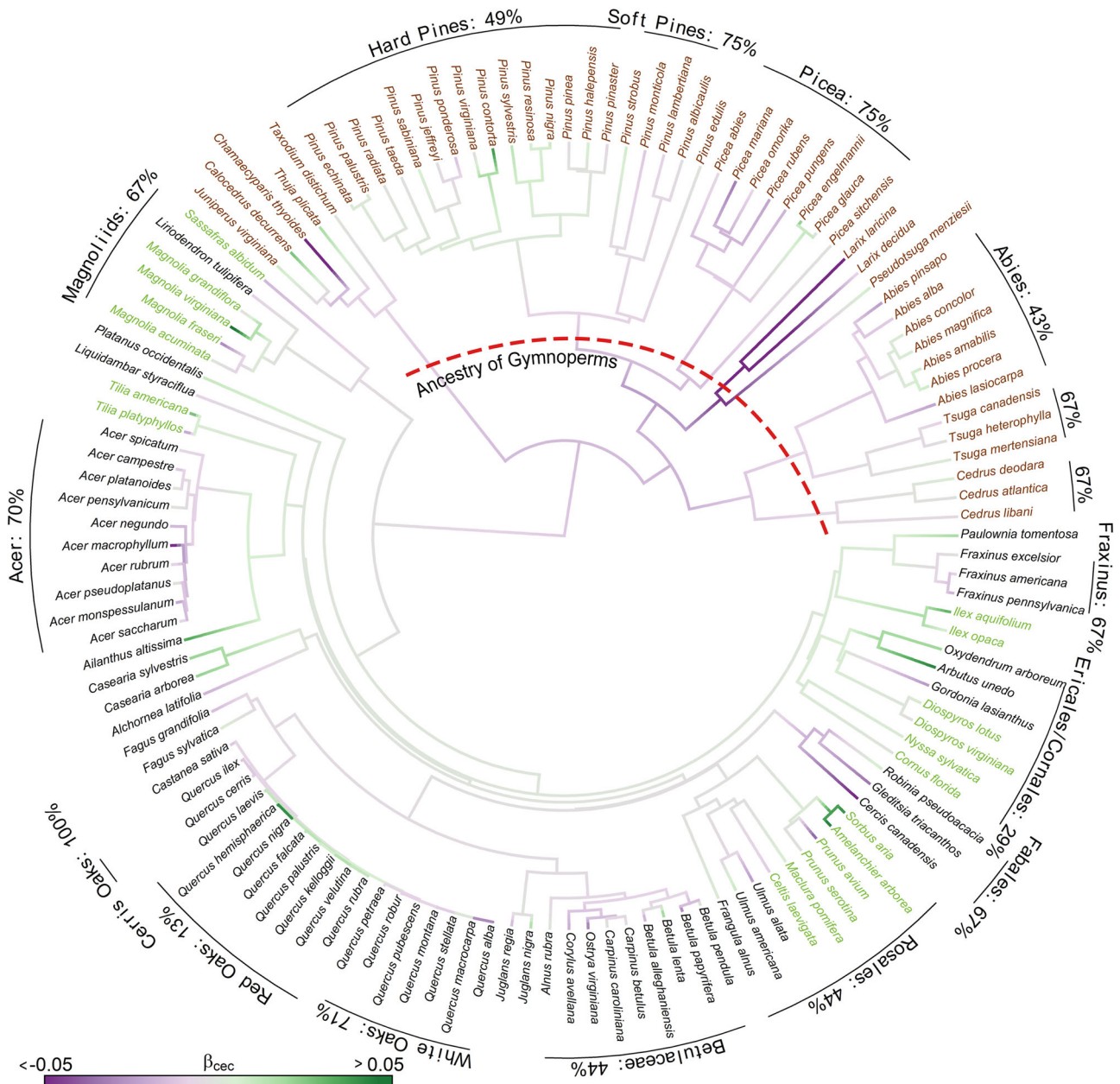

**Fig. 4 Effects of soil fertility on seed production.** Sensitivity of individual standardized production (ISP) to soil fertility based on within-species response to cation exchange capacity ($\beta_{cec}$). Text color follows Fig. 2. Red dashed line indicates the ancestry of gymnosperms. Percentages of species that respond negatively to CEC are labeled for species groups. The analysis includes 141 species that span a sufficient CEC range in the Masting Inference and Forecasting (MASTIF) network to estimate a robust effect. The phylogenetic signal, estimated for 129 species (91% of species) having phylogeny data, is highly significant (*Pagel*'s $\lambda = 0.87$, $p < 0.001$, $n = 129$).

Understanding future forests requires integration of seed production estimates here with drought- and fire-tolerance[88,89], ability to resprout following dieback[90,91], and changes in forest structure and composition[92]. While tree growth and survival have long been quantifiable using traditional methods, they cannot be used to anticipate reforestation potential without knowledge of fecundity. Seed limitation takes on new urgency, as highlighted by fires of uncommon severity in fuels cured by multi-year droughts that leave only remnants of reproductive trees. The relative benefits of size versus numbers of seeds for seedling success in natural regeneration will shift under novel combinations of temperature and moisture[93,94]. The positive effects of nutrient availability on reproduction assumed in current models[36] are sensible for tree growth and NPP[65,95], but not for seed production (Fig. 4). The global relationships quantified from this synthesis bring not only a previously unmeasured dimension of forest response; they also will allow us to leverage existing knowledge of growth and survival with the missing link to regeneration, that of tree fecundity.

## Methods

**Fecundity data.** Fecundity data from the Masting Inference and Forecasting (MASTIF) network[53,96] include two types of raw data, seed traps (ST) and crop counts (CC). Each observation references a tree-year (Fig. 5). Data include longitudinal (repeated) observations on individual trees (99%) and opportunistic observations that come through the iNaturalist project[53]. ST data consist of numbers of seeds that accumulate annually in mapped seed traps on forest inventory plots. A fitted dispersal kernel[53] relates seed counts to mapped trees, accounting for uncertainty in seed transport and Poisson seed counts. Fecundity

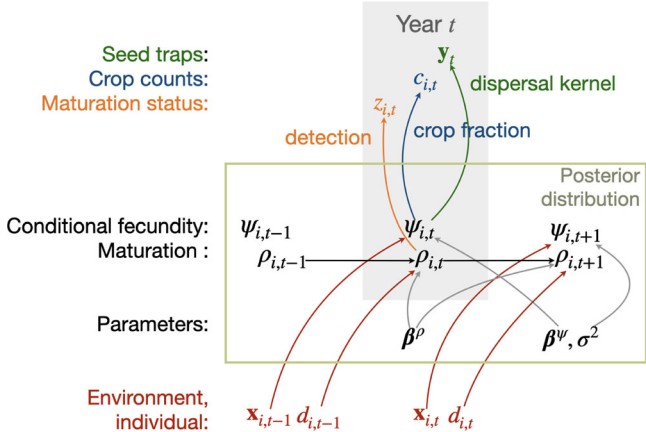

**Fig. 5 The MASTIF model, summarized from Clark et al.[53], includes three levels, observed responses (above), process, and parameters (part of the posterior distribution, middle), and predictors (below).** The data model for observed responses includes uncertainty that comes from seed dispersal (seed traps), the fraction of the crop that can be observed (crop counts), and detection of mature status. The process model describes change in maturation status $\rho_{i,t}$ and, once mature, conditional fecundity $\psi_{i,t}$. Fitted coefficients for conditional fecundity $\beta^\psi$ and maturation probability $\beta^\rho$ describe how predictor variables in red affect maturation and fecundity. Error in the process model is absorbed by process error variance $\sigma^2$. Predictors are held in a design matrix $\mathbf{x}_{i,t}$ for conditional fecundity. Diameter $d_{i,t}$ is the predictor for maturation status. Additional subscripts for location $j$ and species $s$ in the main texts are suppressed to reduce clutter.

modeling (see Fecundity modeling and inference) further accommodates uncertain assignment of seeds to species where seeds are recorded at the genus level[53]. CC data include counts of reproductive structures with estimates of the fraction of the crop observed[53,96]. A beta-binomial distribution accounts for uncertainty in the count and crop-fraction estimates[53,96]. The study includes 12,063,723 tree-years from North America, South and Central America, Europe, Africa, Asia, and Oceania (Supplementary Figure 1) from 751 species. Mature trees for 714 species (95% of 751 species) were used in this analysis (see Fecundity modeling and inference). Data coverage by sites is included in table S2, with sample sizes (tree-years) summarized by species provided in the file **supplementSampleSize.csv**.

**Individual, habitat, and climate covariates**. Covariates were selected based on their ability to explain important variation in the data. In order to test hypotheses related to soil fertility we need covariates that can account for the other sources of variation in data that might obscur hypothesized relationships. Covariates included individual attributes for each tree (diameter and shade classes), climate (temperature and moisture deficit), and habitat conditions (soils and terrain attributes) in Supplementary Table 3. The quadratic term in diameter allows for changes in fecundity response with tree size[21]. Shade classes follow the FIA/NEON classes from 1 (fully-exposed in open space) to 5 (fully-shaded in the understory). Shading quantifies the effects of competition on fecundity[14]. Cation exchange capacity, a widely used indicator for soil fertility[97] at 250-m spatial resolution, was evaluated as a weighted mean based on uncertainty layers from three soil depths (0–5 cm, 5–15 cm, and 15–30 cm) in a global dataset[49] (Supplementary Fig. 1). Slope and aspect quantify effects on solar radiation and drainage on moisture condition. Slope and aspect were obtained from the NASA Shuttle Radar Topography Mission's (SRTM)[98] digital elevation model (DEM) with a spatial resolution of 30 m. The DEM grid was supplemented with the USGS National Elevation Dataset for MASTIF sites that were outside SRTM coverage. Slope and aspect were derived following[99]

$$\mathbf{u} = \begin{bmatrix} u_1 \\ u_2 \\ u_3 \end{bmatrix} = \begin{bmatrix} \sin(\text{slope}) \\ \sin(\text{slope})\sin(\text{aspect}) \\ \sin(\text{slope})\cos(\text{aspect}) \end{bmatrix} \quad (3)$$

Temperature $T$ and moisture deficit $M$ are included in the model as site means and anomalies to account for the effects of both site variation and inter-annual variability[100]. Moisture deficit $M$ summarizes water availability and is obtained from monthly $M = PET - P$ from January to December[101]. $T$ and $M$ were extracted from Terraclimate[102], which provides monthly but spatially coarse data, and CHELSA[103], which is 1 km spatial resolution but is not available after the year 2016. We integrated the two gridded products with local climate monitoring by first using regression to project CHELSA forward based on Terraclimate, with final calibration to sites having local weather data.

**Species traits**. Seed dry mass data were collected by our laboratories and supplemented with the TRY Plant Trait Database[28]. We could not incorporate within-species variation in seed size due to the lack of coverage for this variable across the network. There is some speculation that large seeds are associated with a capacity to generate large crowns in *Pinus pinea*[104] and *Fagus sylvatica*[105]. Mean values per species came from merging our measurements with values in the TRY database. Genus- or family-level means were used where seed mass was missing at the species level. To insure that results are not affected by using genus- and family-level mean, we show the same patterns using only species with species-level seed mass (Supplementary Figs. 5 and 6). Foliar nitrogen and phosphorus concentration, leaf habit, fruit type, and wood density were obtained from the primary literature, TRY, and the BIEN dataset[106].

**Fecundity modeling and inference**. We estimate tree fecundity with a hierarchical Bayes state-space model[53] that incorporates effects of tree attributes with environment to infer effects on maturation and conditional fecundity (Fig. 5). The methodology detailed in[53] builds on models that have provided reliable estimates in many forest types[96,107–115]. The analysis in[53] goes further in showing that the MASTIF model recovers the parameters controlling maturation and fecundity in simulated data, and it accurately predicts the seed-trap data that are used to estimate parameters. To allow for uncertain identification of seeds from trees of the same genus and for dependence within trees over time and between trees, all tree-years of a genus are modeled jointly[53,96]. For each tree $i$ of species $s$ at stand $j$ in year $t$, the expected fecundity is the product of maturation probability and conditional fecundity,

$$E(f_{ijs,t}) = \hat{f}_{ijs,t} = \hat{\rho}_{ijs,t}\hat{\psi}_{ijs,t} \quad (4)$$

Once a tree has been observed to produce seed in a given year $t$, then it is known to be mature in subsequent years $t' > t$. A tree never having been observed to produce seed in the past could be immature or not. Maturation probability $\hat{\rho}_{ijs,t}$ is the probability that an individual is in the mature state,

$$\rho_{ijs,t} \sim Bernoulli\left(\rho_{ijs,t-1} + (1 - \rho_{ijs,t-1})\rho_{ijs,t+1}\Phi(\beta_{s0}^{(\rho)} + \beta_{s1}^{(\rho)}d_{ijs,t})\right) \quad (5)$$

where $\Phi(\cdot)$ represents the standard normal cumulative distribution function (CDF) of the probit model, which is determined by tree diameter $d_{ijs,t}$. Note that coefficients for intercept and diameter $\boldsymbol{\beta}_s^{(\rho)} = [\beta_{s0}^{(\rho)}, \beta_{s1}^{(\rho)}]'$ influence $\rho_{ijs,t}$ only at the transition to the mature state, i.e., $[\rho_{ijs,t}|\rho_{ijs,t-1} = 0, \rho_{ijs,t+1} = 1]$. Conditional fecundity depends on predictors, individual effects, year effects, and error,

$$log(\hat{\psi}_{ijs,t}) = \mathbf{x}'_{ijs,t}\boldsymbol{\beta}_s^{(\psi)} + \beta_{ijs}^{(\psi)} + \gamma_{g[ij]s,t} + \epsilon_{ijs,t} \quad (6)$$

where $\mathbf{x}_{ijs,t}$ is a design matrix holding individual attributes and environmental conditions (see Individual, habitat and climate covariates), and $\boldsymbol{\beta}^{(\psi,x)}$ are fixed-effects coefficients. $\beta_{ijs}^{(\psi)}$ is the random effect for tree $i$ of species $s$ at stand $j$. $\gamma_{g[ij]s,t}$ are year effects that are random across groups $g$ and fixed for year $t$ to account for interannual variation that is not fully captured by climate anomalies. Group membership for year effects ($g[ij]s$) is defined by the species-ecoRegion combination[53]; quasi-synchronicity can vary between species and ecoRegions. There is a noise term $\epsilon_{ijs,t}$. The primary components are summarized in Fig. 5. Variable selection is based on model fit using the Deviance information criterion (DIC) and data coverage, which depends on the range of variation in each predictor and species. Model implementation is open-access with R package `MASTIF`, with full algorithm details provided in[53].

**Individual, species, group, and community seed production**. Individual standardized production (ISP) is expressed as relative to basal area standardized for tree size[21]; it is the product of mass per seed $m_s$ times seeds per tree divided by tree basal area, having units $g\ m^{-2}yr^{-1}$. Each tree-year has an associated posterior mean estimate $\hat{f}_{ijs,t}$. Calculation of ISP combines posterior mean estimates with their uncertainties, as an expectation over years:

$$\text{ISP}_{ijs} = \frac{m_s}{b_{ij}} \times \frac{\sum_t w_{ijs,t}\hat{f}_{ijs,t}}{\sum_t w_{ijs,t}} \quad (7)$$

where $m_s$ is seed mass (g), $b_{ij}$ is basal area ($m^2$), and weight $w_{ijs,t}$ is the inverse of the coefficient of variation (CV) for the estimate,

$$w_{ijs,t} = CV_{ijs,t}^{-1} = \hat{f}_{ijs,t}/s_{ijs,t} \quad (8)$$

and $s_{ijs,t}$ is the standard error of the estimate. The $CV^{-1}$ is used instead of the inverse of variance, because the mean tends to scale with variance. Low values for $\hat{f}_{ijs,t}$ are noisy and less important than high values, which are emphasized by the CV. Estimates of seed number $\hat{f}_{ijs,t}$, its standard error $s_{ijs,t}$, and CV are all dimensionless. All estimates are time averages across annual estimates, so we hereafter omit $yr^{-1}$ from dimensions. Therefore, ISP has the units of $g/m^2$.

Species seed production (SSP) is the expectation taken over individual tree ISP values for a species $s$,

$$SSP_s = \frac{\sum_{ij} w_{ijs} ISP_{ijs}}{\sum_{ij} w_{ijs}} \qquad (9)$$

where $w_{ijs}$ is defined as the inverse $CV_{ijs}$ for individual $i$. Analyses of SSP are done on the proportionate (log) scale to avoid dominance of results by species having highest seed production. Visualizations are based on $\log_{10}$ to facilitate interpretation of scale.

Group seed production (GSP) was evaluated to determine differences between angiosperms versus gymnosperms and between the cone-producing Pinales, large-seeded Fagales, fleshy-fruited members of all groups, and remaining species. For comparisons of species groups $k$, we evaluated means and uncertainties:

$$GSP_k = \frac{\sum_{i,j,s \in k} w_{ijs} ISP_{ijs}}{\sum_{i,j,s \in k} w_{ijs}} \qquad (10)$$

with standard error,

$$SE(GSP_k) = \frac{1}{n_k} \sqrt{\sum_{i,j,s \in k,t} s^2_{ijs,t}} \qquad (11)$$

where $n_k$ is the number of observations included in the estimate. The comparisons between the groups is shown in Supplementary Fig. 3.

To understand the collective impacts of soil fertility on seed production, we calculated community seed production (CSP) by extending tree reproduction to the stand level by summing fecundity over all trees and species and dividing by plot area:

$$CSP_j = \frac{1}{A_j} \sum_{is} ISP_{ijs} \qquad (12)$$

where $A_j$ is plot area ($ha$) for each plot. Note that we did not standardize ISP by basal area to yield instead seeds per area of forest floor. The CSP thus represents stand production, much like NPP represents stand vegetative production. Because calculation of CSP requires plot area, analysis on CSP is based on inventory plots.

**Phylogeny, foliar nutrients, and soil fertility**. Phylogeny was obtained for 482 species (67% of total) from[116]. To determine if SSP can be explained by shared ancestry, we estimated the phylogenetic signal using *Pagel's λ*[117], which ranges from 0 (independence over species) to 1 (related species completely resemble each other). Continuous character mapping was used to visualize the phylogenetic signal in Fig. 2 using R package `phytools`[118]. We explored the linkage between foliar nutrients and species difference in seed production using the phylogenetic regression to account for species relatedness, where the correlation between species increases proportionally with the branch length (i.e., from the root to the most recent common ancestor) in the phylogeny[119,120]. Phylogenetic regression was implemented using R package `phylolm`[121] for 427 species that has foliar nutrients data (89% of the 482 species with phylogeny).

We estimated within-species response to CEC, that is, $\beta_{cec}$ in Eq. (6), for 141 species that has sufficient range of variation across the soil fertility (CEC) gradient; $\beta_{cec}$ coefficients cannot be estimated for species that are limited to one or a few sites. We quantified phylogenetic coherence in $\beta_{cec}$ (Fig. 4) between species following the same procedure used for SSP. To determine how the ability to estimate $\beta_{cec}$ depended on the range in CEC over which it was observed, we plotted predicted seed production against CEC while holding all other covariates constant (i.e., half of maximum diameter, medium shade, and mean temperature and moisture in Supplementary Fig. 4). This figure demonstrates the large differences in both directions and magnitudes of $\beta_{cec}$ along the soil fertility gradients where the species have been observed.

Finally, we evaluated the effects of soil fertility on stand productivity (i.e., CSP) using regression. The analysis is restricted to sites having all trees sampled within a known area, which is true for all inventory studies with seed-traps.

**Calculation of seed number per 1 g**. Provided the estimate of $\alpha$ does not have large error, the expected seeds per basal area at seed mass = 1 g for the $\log_{10}$-$\log_{10}$ regression is approximated with two terms from the Taylor expansion

$$E(10^\alpha) = 10^{\hat{\alpha}} \left( 1 + \frac{s^2_\alpha l^2}{2} \right) \pm 10^{\hat{\alpha}} s_\alpha l \qquad (13)$$

where $\hat{\alpha} = 4.29$ and $s_\alpha = 0.065$ are the estimate and standard error for parameter $\alpha$, and the constant $l = \log(10)$. For this model $\hat{\alpha} \gg s_\alpha$, this approximation is confident at $19,720 \pm 2,920$. For a tree of given size, its seed production is $19,720 \times$ its basal area.

**Seed size-number trade-offs within orders**. We have included taxonomic rank (i.e., order) as a factor level, including an interaction with seed size, in the regression. We found that SSP was not constrained by the strict 1:1 trade-off within each order (Supplementary Table 4).

**Reporting summary**. Further information on research design is available in the Nature Research Reporting Summary linked to this article.

## Data availability

Seed production data are available at the Duke Data Repository https://doi.org/10.7924/r4348ph5t[122]. Species traits are downloaded from TRY Plant Trait database at https://www.try-db.org/TryWeb/Home.php[28]. Cation exchange capacity data are obtained at https://soilgrids.org/[49]. Climate data are extracted from Terraclimate at http://www.climatologylab.org/[102] and CHELSA at https://chelsa-climate.org/[103]. Elevation data are obtained from SRTM at https://srtm.csi.cgiar.org/ and USGS National Elevation Dataset at https://ned.usgs.gov/.

## Code availability

R statistical software v4.0.2 was used in this work. All analyses used published R packages, with details stated in the section Methods. MASTIF includes code in R and C++, which is published on CRAN at https://cran.r-project.org/web/packages/mastif/index.html.

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

## Acknowledgements

For access to sites and logistical support we thank the National Ecological Observatory Network (NEON). The project has been funded continuously since 1992 by National Science Foundation grants to J.S.C, most recently DEB-1754443, and by the Belmont Forum (1854976), NASA (AIST16-0052, AIST18-0063), and the Programme d'Investissement d'Avenir under project FORBIC (18-MPGA-0004)(*Make Our Planet Great Again*). Puerto Rico data were funded by NSF grants to M.U., most recently, DEB 0963447 and LTREB 11222325. Data from the Andes Biodiversity and Ecosystem Research Group were funded by the Gordon and Betty Moore Foundation and NSF LTREB 1754647 to M.S. Additional funding to M.Z. came from the W.Szafer Institute of Botany of the Polish Academy of Sciences and the Polish National Science Foundation (2019/33/B/NZ8/0134). M.B. was supported by the Polish National Science Centre grant no. 2019/35/D/NZ8/00050, and Polish National Agency for Academic Exchange Bekker programme PPN/BEK/2020/1/00009/U/00001. J.M.L. was supported by NSF grant DEB 1745496. Jerry Franklin's data remain accessible through NSF LTER DEB-1440409. USDA Forest Service and USGS research was funded by those agencies. We thanked Walt Koenig for providing data. We thanked Christopher W. Woodall for comments on the manuscript. Any use of trade, firm, or product names does not imply endorsement by the U.S. Government.

## Author contributions

J.S.C and T.Q. designed the study, performed analyses, and wrote the paper, J.S.C. compiled the MASTIF data and wrote the MASTIF model and software, B.C., V.J, and G.K. co-wrote the paper. T.Q., R.A., M.A., D.A., Y.B., R.B., D.B., M.B., T.B., R.B., D.C.B., T.C., R.C., J.J.C., C.C., N.L.C., B.C., F.C., T.C., A.J.D., E.D., H.D., N.D., S.D., M.D., S.D., L.D., J.E., T.J.F., W.F., C.A.G., G.S.G., G.G., C.H.G., Q.G., A.H., A.H., Q.H., J.H., K.H., I.I., J.F.J., V.J., D.K., C.L.K., T.K., J.M.K., R.K.K., G.K., J.G.L., J.M.L., M.L., F.L., T.L., J.L., J.A.L., D.M., E.J.M., C.M.M., E.M., R.M., J.A.M., T.A.N., K.N., J.O., R.P., I.S.P., I.M.P., L.P., J.P., R.P., M.D.R., C.D.R., K.C.R., F.R., J.D.S., C.L.S., W.H.S., H.S., B.S., S.S., M.S., M.A.S., N.L.S., J.N.S., I.S., S.S., J.J.S., M.S., P.A.T., M.U., G.V., T.T.V., A.V.W., T.G.W., A.P.W., B.W., S.J.W., K.Z., J.K.Z., R.Z., M.Z., and J.S.C contributed data and revised the paper.

## Competing interests

The authors declare no competing interests.

## Additional information

Tong Qiu [1], Robert Andrus[2], Marie-Claire Aravena [3], Davide Ascoli[4], Yves Bergeron [5], Roberta Berretti[4], Daniel Berveiller[6], Michal Bogdziewicz [7], Thomas Boivin[8], Raul Bonal[9], Don C. Bragg[10], Thomas Caignard[11], Rafael Calama[12], J. Julio Camarero [13], Chia-Hao Chang-Yang [14], Natalie L. Cleavitt[15], Benoit Courbaud[16], Francois Courbet[8], Thomas Curt[17], Adrian J. Das [18], Evangelia Daskalakou [19], Hendrik Davi[8], Nicolas Delpierre[6], Sylvain Delzon[11], Michael Dietze [20], Sergio Donoso Calderon[3], Laurent Dormont[21], Josep Espelta [22], Timothy J. Fahey[15], William Farfan-Rios [23], Catherine A. Gehring[24], Gregory S. Gilbert [25], Georg Gratzer[26], Cathryn H. Greenberg[27], Qinfeng Guo [28], Andrew Hacket-Pain [29], Arndt Hampe[11], Qingmin Han [30], Janneke Hille Ris Lambers[31], Kazuhiko Hoshizaki [32], Ines Ibanez [33], Jill F. Johnstone [34], Valentin Journé [16], Daisuke Kabeya[30], Christopher L. Kilner [1], Thomas Kitzberger[35], Johannes M. H. Knops[36], Richard K. Kobe[37], Georges Kunstler[16], Jonathan G. A. Lageard [38], Jalene M. LaMontagne [39], Mateusz Ledwon[40], Francois Lefevre [8], Theodor Leininger[41], Jean-Marc Limousin[42], James A. Lutz [43], Diana Macias[44], Eliot J. B. McIntire[45], Christopher M. Moore[46], Emily Moran [47], Renzo Motta[4], Jonathan A. Myers [48], Thomas A. Nagel[49], Kyotaro Noguchi[50], Jean-Marc Ourcival[42], Robert Parmenter [51], Ian S. Pearse[52], Ignacio M. Perez-Ramos[53], Lukasz Piechnik[54], John Poulsen[1], Renata Poulton-Kamakura[1], Miranda D. Redmond[55], Chantal D. Reid[1], Kyle C. Rodman [56], Francisco Rodriguez-Sanchez[57], Javier D. Sanguinetti[58], C. Lane Scher[1], William H. Schlesinger[1], Harald Schmidt Van Marle[3], Barbara Seget[54], Shubhi Sharma[1], Miles Silman[59], Michael A. Steele[60], Nathan L. Stephenson [18], Jacob N. Straub[61], I-Fang Sun [62], Samantha Sutton[1], Jennifer J. Swenson[1], Margaret Swift[1], Peter A. Thomas [63], Maria Uriarte [64], Giorgio Vacchiano [65], Thomas T. Veblen [2], Amy V. Whipple[24], Thomas G. Whitham[24], Andreas P. Wion[66], Boyd Wright[67], S. Joseph Wright [68], Kai Zhu [25], Jess K. Zimmerman[69], Roman Zlotin[70], Magdalena Zywiec[54] & James S. Clark [1,16] ✉

[1]Nicholas School of the Environment, Duke University, Durham, NC 27708, USA. [2]Department of Geography, University of Colorado Boulder, Boulder, CO 80309, USA. [3]Universidad de Chile, Facultad de Ciencias Forestales y de la Conservacion de la Naturaleza (FCFCN), La Pintana 8820808 Santiago, Chile. [4]Department of Agriculture, Forest and Food Sciences, University of Torino, 10095 Grugliasco, TO, Italy. [5]Forest Research Institute, University of Quebec in Abitibi-Temiscamingue, Rouyn-Noranda, QC J9X 5E4, Canada. [6]Universite Paris-Saclay, Centre national de la recherche scientifique, AgroParisTech, Ecologie Systematique et Evolution, 91405 Orsay, France. [7]Department of Systematic Zoology, Faculty of Biology, Adam Mickiewicz University, Umultowska 89, 61-614 Poznan, Poland. [8]Institut National de Recherche pour Agriculture, Alimentation et Environnement (INRAE), Ecologie des Forets Mediterranennes, 84000 Avignon, France. [9]Department of Biodiversity, Ecology and Evolution, Complutense University of Madrid, 28040 Madrid, Spain. [10]USDA Forest Service, Southern Research Station, Monticello, AR 71656, USA. [11]Universite Bordeaux, Institut National de Recherche pour Agriculture, Alimentation et Environnement (INRAE), Biodiversity, Genes, and Communities (BIOGECO), 33615 Pessac, France. [12]Centro de Investigacion Forestal - Instituto Nacional de Investigacion y Tecnologia Agraria y Alimentaria (INIA-CIFOR), 28040 Madrid, Spain. [13]Instituto Pirenaico de Ecologla, Consejo Superior de Investigaciones Cientificas (IPE-CSIC), 50059 Zaragoza, Spain. [14]Department of Biological Sciences, National Sun Yat-sen University, Kaohsiung 80424, Taiwan. [15]Natural Resources, Cornell University, Ithaca, NY 14853, USA. [16]Universite Grenoble Alpes, Institut National de Recherche pour Agriculture, Alimentation et Environnement (INRAE), Laboratoire EcoSystemes et Societes En Montagne (LESSEM), 38402 St. Martin-d'Heres, France. [17]Aix Marseille universite, Institut National de Recherche pour Agriculture, Alimentation et Environnement (INRAE), 13182 Aix-en-Provence, France. [18]USGS Western Ecological Research Center, Three Rivers, CA 93271, USA. [19]Institute of Mediterranean and Forest Ecosystems, Athens, Greece. [20]Earth and Environment, Boston University, Boston, MA 02215, USA. [21]Centre d'Ecologie Fonctionnelle et Evolutive (CEFE), Centre National de la Recherche Scientifique (CNRS), 34293 Montpellier, France. [22]Centre de Recerca Ecologica i Aplicacions Forestals (CREAF), Bellaterra, Catalunya 08193, Spain. [23]Washington University in Saint Louis, Center for Conservation and Sustainable Development, Missouri Botanical Garden, St. Louis, MO 63110, USA. [24]Department of Biological Sciences, Northern Arizona University, Flagstaff, AZ 86011, USA. [25]Department of Environmental Studies, University of California, Santa Cruz, CA 95064, USA. [26]Institute of Forest Ecology, Peter-Jordan-Strasse 82, 1190 Wien, Austria. [27]Bent Creek Experimental Forest, USDA Forest Service, Asheville, NC 28801, USA. [28]Eastern Forest Environmental Threat Assessment Center, USDA Forest Service, Southern Research Station, Research Triangle Park, NC 27709, USA. [29]Department of Geography and Planning, School of Environmental Sciences, University of Liverpool, Liverpool, UK. [30]Department of Plant Ecology Forestry and Forest Products Research Institute (FFPRI), Tsukuba, Ibaraki 305-8687, Japan. [31]Department of Environmental Systems Science, ETH Zurich, Zurich 8092, Switzerland. [32]Department of Biological Environment, Akita Prefectural University, Akita 010-0195, Japan. [33]School for Environment and Sustainability, University of Michigan, Ann Arbor, MI 48109, USA. [34]Institute of Arctic Biology, University of Alaska, Fairbanks, AK 99700, USA. [35]Department of Ecology, Instituto de Investigaciones en Biodiversidad y Medioambiente (Consejo Nacional de Investigaciones Cientificas y Tecnicas - Universidad Nacional del

Comahue), Quintral 1250, 8400 Bariloche, Argentina. [36]Health and Environmental Sciences Department, Xian Jiaotong-Liverpool University, Suzhou 215123, China. [37]Department of Plant Biology, Program in Ecology, Evolutionary Biology, and Behavior, Michigan State University, East Lansing, MI 48824, USA. [38]Department of Natural Sciences, Manchester Metropolitan University, Manchester M1 5GD, UK. [39]Department of Biological Sciences, DePaul University, Chicago, IL 60614, USA. [40]Institute of Systematics and Evolution of Animals, Polish Academy of Sciences, Slawkowska 17, 31-016 Krakow, Poland. [41]USDA, Forest Service, Southern Research Station, PO Box 227, Stoneville, MS 38776, USA. [42]CEFE, Univ Montpellier, CNRS, EPHE, IRD, 1919 route de Mende, 34293 Montpellier Cedex 5, France. [43]Department of Wildland Resources, and the Ecology Center, Utah State University, Logan, UT 84322, USA. [44]Department of Biology, University of New Mexico, Albuquerque, NM 87131, USA. [45]Pacific Forestry Centre, Victoria, BC V8Z 1M5, Canada. [46]Department of Biology, Colby College, Waterville, ME 04901, USA. [47]School of Natural Sciences, UC Merced, Merced, CA 95343, USA. [48]Department of Biology, Washington University in St. Louis, St. Louis, MO, USA. [49]Department of forestry and renewable forest resources, Biotechnical Faculty, University of Ljubljana, Ljubljana, Slovenia. [50]Tohoku Research Center, Forestry and Forest Products Research Institute, Morioka, Iwate 020-0123, Japan. [51]Valles Caldera National Preserve, National Park Service, Jemez Springs, NM 87025, USA. [52]Fort Collins Science Center, 2150 Centre Avenue Bldg C, Fort Collins, CO 80526, USA. [53]Inst. de Recursos Naturales y Agrobiologia de Sevilla, Consejo Superior de Investigaciones Cientificas (IRNAS-CSIC), Seville, Andalucia, Spain. [54]W. Szafer Institute of Botany, Polish Academy of Sciences, Lubicz 46, 31-512 Krakow, Poland. [55]Department of Forest and Rangeland Stewardship, COlorado State University, Fort COllins, CO, USA. [56]Department of Forest and Wildlife Ecology, University of Wisconsin-Madison, Madison, WI 53706, USA. [57]Department of Biologia Vegetal y Ecologia, Universidad de Sevilla, 41012 Sevilla, Spain. [58]Bilogo Dpto. Conservacin y Manejo Parque Nacional Lanin Elordi y Perito Moreno, 8370 San Marten de los Andes Neuqun, Argentina. [59]Department of Biology, Wake Forest University, 1834 Wake Forest Rd, Winston-Salem, NC 27106, USA . [60]Department of Biology, Wilkes University, 84 West South Street, Wilkes-Barre, PA 18766, USA. [61]Department of Environmental Science and Ecology, State University of New York-Brockport, Brockport, NY 14420, USA. [62]Center for Interdisciplinary Research on Ecology and Sustainability, College of Environmental Studies, National Dong Hwa University, Hualien, Taiwan. [63]School of Life Sciences, Keele University, Staffordshire ST5 5BG, UK. [64]Department of Ecology, Evolution and Environmental Biology, Columbia University, 1113 Schermerhorn Ext., 1200 Amsterdam Ave., New York, NY 10027, USA. [65]Department of Agricultural and Environmental Sciences - Production, Territory, Agroenergy (DISAA), University of Milan, 20133 Milano, Italy. [66]Department of Forest and Rangeland Stewardship, Colorado State University, Fort Collins, CO 80523, USA. [67]Botany, School of Environmental and Rural Science, University of New England, Armidale, NSW 2350, Australia. [68]Smithsonian Tropical Research Institute, Apartado 0843n03092, Balboa, Republic of Panama. [69]Department of Environmental Sciences, University of Puerto Rico, Rio Piedras, PR 00936, USA. [70]Geography Department and Russian and East European Institute, Bloomington, IN 47405, USA. ✉email: jimclark@duke.edu

