## [Peer Review File · Nature Communications]

Reviewers' Comments:

Reviewer #1:

Remarks to the Author:

The study is a substantial piece of work. The research questions are relevant and clearly hypothesised, and authors sought to answer them with a substantial dataset covering many species and lots of observations. I have some minor methodological concerns, which authors need to address. So I would recommend revision of the study to consider the following points.

Main comments:

The core of the study hinges in the individual seed production (ISP) and the (expected) species seed productivity (SSP). From Equations 1&2, we can see that both quantities are inferred from seed mass. Specifically, ISP is the product of seed mass per the mean seed numbers per tree basal area, while SSP is the product of seed mass for the expectation of mean seed numbers per basal area. While the mean number of seeds per tree basal area are directly retrieved (i.e., measured in case of traps, estimated in case of visual counts) from MASTIF, seed mass is not directly available from this dataset.

By taking the seed mass from TRY (or primary sources, which should be clearly reported in the manuscript), authors implicitly assumed that individual seed mass is 'constant' (or rather species-specific) and does not scale with tree size. However, seed mass itself may vary with tree size. Larger trees have typically larger crowns, which not solely have larger 'space' to hosts more abundant seeds, but may allow production of larger seeds. Although poorly documented, there are some examples of such phenomenon in species like stone pine (*Pinus pinea*; <https://doi.org/10.3390/rs12010173>) and beech(<https://doi.org/10.15177/seefor.18-06>) and it is interesting that for these species some pruning and thinning methods are meant to promote larger seed yields via promoting larger individual crowns. Larger seeds within-species may be likely also related to an higher seed germinability, which in turns may influence the tree regeneration capacity.

I understand the simplification needed for a global synthesis, but of course the matter deserves future investigations. So I suggest author to acknowledge the limitation in their framework when assuming seed mass in their calculations.

I have just minor questions, mainly clarification needed:

Equation 6: didn't understand what are the units and quantities used in the weighting factor (Eq. 7) in ISP. Please clarify.

Equation 9 CSP is plot-level estimate based on traps or visual counts. If I have says 25 traps each 1 square meter in surface, which are 'representative' or 1 ha plot, I would have $\text{Sum}(\text{ISP})/10000$, but it is not per hectare estimate, as I should divide for the total surface (i.e., is not the sum(ISP) but the mean) or I am getting wrong?

It would be interesting to see how was the covariance between genus or higher taxonomic orders in the linear equation in 2a made by ancova or some post-hoc tests.

Line 2-3 & 172-177. The core of the investigation is in the potential of seed production, which is strictly not related to germination, although there is a likely expected positive relationship between seed production and seed germination - but is not always the case.

So I would revise these sentences.

Indeed, while it is reasonable to assume that the rooted existence of plants have (among many other vegetation strategies) promoted different reproductive structures to adapt vegetation to environment, the link with regeneration failure is not strictly justified by the reproductive constraints, nor artificial regeneration will be necessarily a 'reliable' reliance solution. Plant strategies like resprouting, changing community species composition between herb-shrub-woody, ruderal/climax species or alternative dispersal are some of the processes not strictly mediated by reproduction which are in the portfolio of vegetation responses to future disturbances.

Reviewer #2:

Remarks to the Author:

Comments for authors

I read the current ms with the greatest interest. This ms uses an impressive, global tree database in which seed production data based on ca. 12.1 M observations on 714 tree species are combined, to estimate the relationship between total seed output of trees, their seed size and number and to link these fecundity variables to basal area, nutrient status and phylogeny. Also the impact of several environmental variables (incl. soil CEC) and plant traits (incl. foliar nutrient concentrations) on these relationships was studied. The most important finding the authors claim is that there is no 1:1 trade-off between seed size (mass) and number, at least not across species. Instead, tree species that produce large seeds also produce relatively many seeds, so that their total seed output (g seed produced per unit basal area) is generally higher than that for tree species that produce smaller seeds. A secondary finding is that there is a strong phylogenetic signal in total seed output, with gymnosperms producing slightly lower seed outputs than angiosperms. Finally, the authors detect a relation between foliar phosphorus content and seed output.

While the large dataset used for the analyses is certainly impressive, I find the current ms to be extremely dense and much important information is missing.

From an ecological perspective, it is not clear to me:

1) Why the competition-colonization trade-off is not included in the study. L34-35: "Limits to SSP [total tree seed output] could depend on genetics, resource access, and cost of auxiliary structures associated with seed production". Another major limit is likely to be the life history strategy of a tree species, particularly along the competition-colonization trade-off (i.e., species either invest more in fast growth and high reproductive output, or more in slow growth and high wood density, competitive dominance and low reproductive output). This may be a major driver of species differences in SSP. It could be easily taken into account by adding wood density as variable in the model. Wood density is a widely available tree trait and highly representative of species life history strategy. Why is this not included?

2) What exactly is the expectation for hypothesis B (62-63)? That species SSP varies with soil fertility, but how? Do the authors expect an increase, a decrease, or a unimodal relation? Why? If this is indeed a genotype x environment response, how is the genotypical variation addressed in the study? I cannot assess from the model whether the slopes of the relation between SSP and CEC are allowed to vary between individuals – is this the case? What is the expectation for the differences between species? That species that produce auxiliary structures such as fruits and cones have lower SSP? Then why is this not tested? I see an analysis comparing angiosperms to gymnosperms (which have cones) but an analysis comparing fruit-bearing angiosperms to non-fruit-bearing angiosperms is missing.

3) Similarly, what exactly is the expectation for hypothesis C (L64)? That species sensitivities to nutrient availability drive the relation between community SP (CSP) and CEC? Or that species turnover drives this relation? Why?

4) In relation to point 1, in the Discussion (L123-124) it is stated: "Results show that the increased seed-mass production realized by species that invest in large seeds is mediated by resource access and the cost of fruiting structures". I don't see the evidence for the cost of fruiting structures; a test of impact of fruits on SSP is missing.

5) In the Introduction and (particularly) Discussion, I find the link between SSP and tree regeneration and future recovery after disturbances such as fire a bit far-fetched. Tree SSP does not have much implications for regeneration: if it did, only the tree species with high SSP would survive. Instead, the current ms nicely shows (and I think for the first time on such a scale) that tree SSP varies across several orders of magnitude, thereby emphasizing how widely different the regeneration strategies across tree species are. All these tree species survive in their own habitats, suggesting that a very wide range of different regeneration strategies are possible, depending on life history strategy, habitat type and (probably) disturbance regime. Links to the latter remain

speculative however, as the current ms did not evaluate this.

From a methodological perspective, I am missing important pieces of information. I find the Methods part particularly sparse of information. The section limits may not allow sufficient word space to provide all relevant details, but in that case I suggest to provide this information in Online Material. In particular, it is unclear to me:

6) Why the selected covariates have been selected. Some of them seem part of hypotheses (e.g., CEC) while most others do not. What was the reason behind including them? I can imagine some (e.g. shade class) can account for differences in SSP, but for others (e.g. slope) I find it hard to imagine such a relation. Please explain.

7) How were seed crop counts and seed trap data analyzed? Were they analyzed separately, or combined? If the latter, how?

8) If the latter, how were seed trap data converted to match crop counts? Also, how were seed trap data allocated to individual trees (needed to obtain data on basal area)?

9) L234-235 "Maturation probability rho accounts for the immature state (for small trees) and failed crop in larger trees": How? Based on what data?

Finally, I have one important methodological comment:

L218-219: "Genus or family-level means were used where seed mass is missing at species level" this is indeed frequently done but my experience is that for seed size this method is highly unreliable. Seed mass differences of several orders of magnitude exist among species within the same family, and within the same genus this is even possible. By replacing species with lacking data by inappropriate estimates known to vary so widely within genera and families, data reliability is unnecessarily reduced. I strongly recommend to remove species for which seed mass is unknown at the species level from the analysis. I am expecting that for most tree species, seed mass is known as this is one of the most widely available seed traits, so that this will not greatly affect the number of tree species to be included in the analysis.

Some more detailed comments:

L27: alpha is the log10 (SSP at seed mass equal to 1 g), I presume?

Attached datafile: I only find data for 646 species, the other species contain zeroes for both crop counts and trapped seeds. Where do the data on the remaining species in the analysis (714-646=68 species) come from?

Reviewer #3:

Remarks to the Author:

This is a very interesting and complex paper. I do not completely understand all the details of your analyses, but overall I am impressed and have nothing in the way of criticism. The implications are clear: there is little to no evidence for a major tradeoff in seed production vs. seed size. The biggest criticism I have is this: now what? I recognize that part of this limitation may be constraints imposed by the journal, which I am not very familiar with, but ...

Imagine I have a student interested in this problem. Clearly, I should not advise him to pursue any kind of traditional tradeoff to explain variation in seed production, although I do note that an important factor you have not been able to consider is spatial variation: big seeds also need to be moved away from the parent. Where do you think we should go instead? I wish you had attempted this kind of analysis.

Overall, very well done. Congratulations.

We appreciated the efforts and comments from all reviewers, which have greatly improved the manuscript. Below please find our point-to-point response.

REVIEWER COMMENTS

Reviewer #1 (Remarks to the Author):

The study is a substantial piece of work. The research questions are relevant and clearly hypothesised, and authors sought to answer them with a substantial dataset covering many species and lots of observations. I have some minor methodological concerns, which authors need to address. So I would recommend revision of the study to consider the following points.

Main comments:

The core of the study hinges in the individual seed production (ISP) and the (expected) species seed productivity (SSP). From Equations 1&2, we can see that both quantities are inferred from seed mass. Specifically, ISP is the product of seed mass per the mean seed numbers per tree basal area, while SSP is the product of seed mass for the expectation of mean seed numbers per basal area. While the mean number of seeds per tree basal area are directly retrieved (i.e., measured in case of traps, estimated in case of visual counts) from MASTIF, seed mass is not directly available from this dataset.

By taking the seed mass from TRY (or primary sources, which should be clearly reported in the manuscript), authors implicitly assumed that individual seed mass is 'constant' (or rather species-specific) and does not scale with tree size. However, seed mass itself may vary with tree size. Larger trees have typically larger crowns, which not solely have larger 'space' to hosts more abundant seeds, but may allow production of larger seeds. Although poorly documented, there are some examples of such phenomenon in species like stone pine (*Pinus pinea*; <https://doi.org/10.3390/rs12010173>) and beech (<https://doi.org/10.15177/seefor.18-06>) and it is interesting that for these species some pruning and thinning methods are meant to promote larger seed yields via promoting larger individual crowns. Larger seeds within-species may be likely also related to an higher seed germinability, which in turns may influence the tree regeneration capacity.

I understand the simplification needed for a global synthesis, but of course the matter deserves future investigations. So I suggest author to acknowledge the limitation in their framework when assuming seed mass in their calculations.

We appreciate this point and have acknowledged this possibility in the revision. We agree that seed-mass variation within species or even within trees is poorly documented. As the reviewer points out, mean values for species are used as the best that can be done for this and other traits in the large trait literature. Having said that, the effect of crown area on size of the crop (seed number) is clear from ours and others going back at least to 2004 (Clark et al, *Ecol Monogr*). We have expanded the text in lines 162-165. We have cited the effect of crown status and the potential connection between crown size and seed size mentioned by this reviewer in lines 254-256. Finally, we clarified the primary sources on lines 253-254.

I have just minor questions, mainly clarification needed:

Equation 6: didn't understand what are the units and quantities used in the weighting factor (Eq. 7) in ISP. Please clarify.

The weighting factor is the inverse of coefficient of variation and thus is dimensionless. It is the ratio of log fecundity and its standard error (both dimensionless). We have fully revised this section, including clarifying units on lines 295-298.

Equation 9 CSP is plot-level estimate based on traps or visual counts. If I have says 25 traps each 1 square meter in surface, which are 'representative' or 1 ha plot, I would have Sum(ISP)/10000 , but it is not per hectare estimate, as I should divide for the total surface (i.e., is not the sum(ISP) but the mean) or I am getting wrong?

We see how this could be confusing given that estimates like the one described here are not uncommon in the literature. On lines 70, we emphasize that we do not obtain CSP as an extrapolation of counts in seed traps. That practice would generate huge error due to the precise locations of a small number of seed traps. The only reliable estimate of seeds production per ha has to sum the production over all trees in the plot and divide by the area of the plot. We have clarified this on lines 70-75.

It would be interesting to see how was the covariance between genus or higher taxonomic orders in the linear equation in 2a made by ancova or some post-hoc tests.

We have included order as a factor level, including an interaction with seed size, in the regression, which is now included as table S4. These slopes are positive with the exception of Magnoliids, which is poorly represented and not significant.

Line 2-3 & 172-177. The core of the investigation is in the potential of seed production, which is strictly not related to germination, although there is a likely expected positive relationship between seed production and seed germination - but is not always the case.

So I would revise these sentences.

Indeed, while it is reasonable to assume that the rooted existence of plants have (among many other vegetation strategies) promoted different reproductive structures to adapt vegetation to environment, the link with regeneration failure is not strictly justified by the reproductive constraints, nor artificial regeneration will be necessarily a 'reliable' reliance solution. Plant strategies like resprouting, changing community species composition between herb-shrub-woody, ruderal/climax species or alternative dispersal are some of the processes not strictly mediated by reproduction which are in the portfolio of vegetation responses to future disturbances.

We're not sure if the reviewer is disagreeing that increasing the number of seeds increases the chances for successful germination. We fully agree that vegetative reproduction is an important reproductive strategy for many species in many environments. Line 5 says this, which does not seem at odds with the reviewer comment:

"Species capable of vegetative regrowth will contribute to reforestation [9], but colonization and seedling success in many landscapes will depend on numbers and sizes of seeds from surviving trees."

We have revised and expanded our statements about the relative importance of regeneration by seed, both naturally and in forest practice, are supported by the literature, with many citations (beginning on line 197). The final paragraph on line 205 is also supported by the literature and the results of this study. Still, we may not fully understand this comment, but believe that the additional editing may help.

Reviewer #2 (Remarks to the Author):

Comments for authors

I read the current ms with the greatest interest. This ms uses an impressive, global tree database in which seed production data based on ca. 12.1 M observations on 714 tree species are combined, to estimate the relationship between total seed output of trees, their seed size and number and to link these fecundity variables to basal area, nutrient status and phylogeny. Also the impact of several environmental variables (incl. soil CEC) and plant traits (incl. foliar nutrient concentrations) on these relationships was studied. The most important finding the authors claim is that there is no 1:1 trade-off between seed size (mass) and number, at least not across species. Instead, tree species that produce large seeds also produce relatively many seeds, so that their total seed output (g seed produced per unit basal area) is generally higher than that for tree species that produce smaller seeds. A secondary finding is that there is a strong phylogenetic signal in total seed output, with gymnosperms producing slightly lower seed outputs than angiosperms. Finally, the authors detect a relation between foliar phosphorus content and seed output.

While the large dataset used for the analyses is certainly impressive, I find the current ms to be extremely dense and much important information is missing.

From an ecological perspective, it is not clear to me:

1) Why the competition-colonization trade-off is not included in the study. L34-35: "Limits to SSP [total tree seed output] could depend on genetics, resource access, and cost of auxiliary structures associated with seed production". Another major limit is likely to be the life history strategy of a tree species, particularly along the competition-colonization trade-off (i.e., species either invest more in fast growth and high reproductive output, or more in slow growth and high wood density, competitive dominance and low reproductive output). This may be a major driver of species differences in SSP. It could be easily taken into account by adding wood density as variable in the model. Wood density is a widely available tree trait and highly representative of species life history strategy. Why is this not included?

We agree that it is relevant to mention the colonization-competition concept. We have added it to the Introduction (line 36-39), and we included wood density in the analysis (line 259-261). We found that wood density does not explain variation in SSP. We added this finding to the Results (line 122) and added the table that summarizes the coefficients to the supplements (table S5).

2) What exactly is the expectation for hypothesis B (62-63)? That species SSP varies with soil fertility, but how? Do the authors expect an increase, a decrease, or a unimodal relation? Why? If this is indeed a genotype x environment response, how is the genotypical variation addressed in the study? I cannot assess from the model whether the slopes of the relation between SSP and CEC are allowed to vary between individuals – is this the case? What is the expectation for

the differences between species? That species that produce auxiliary structures such as fruits and cones have lower SSP? Then why is this not tested? I see an analysis comparing angiosperms to gymnosperms (which have cones) but an analysis comparing fruit-bearing angiosperms to non-fruit-bearing angiosperms is missing.

3) Similarly, what exactly is the expectation for hypothesis C (L64)? That species sensitivities to nutrient availability drive the relation between community SP (CSP) and CEC? Or that species turnover drives this relation? Why?

We addressed 2) and 3) together. Because some readers might not connect the hypotheses with the preceding paragraphs where the alternative hypotheses are explained, we have revised those paragraphs (starting on line 40) to emphasize how current understanding supports these alternatives. For hypothesis A, alternatives are explained starting on line 40. For hypothesis B, alternatives are explained starting on line 48. For hypothesis C, alternatives are explained starting on line 60. For hypothesis D, alternatives are explained starting on line 68. We believe that the revision and reorganization will alleviate any confusion on the alternative outcomes for each hypothesis.

The model does not include individual variation in response to CEC, because each individual occupies one site with one CEC value. An individual random effect is included in the model (line 277), because individuals are observed repeatedly. However, random slopes can only be estimated for variables that vary within an individual.

4) In relation to point 1, in the Discussion (L123-124) it is stated: “Results show that the increased seed-mass production realized by species that invest in large seeds is mediated by resource access and the cost of fruiting structures”. I don’t see the evidence for the cost of fruiting structures; a test of impact of fruits on SSP is missing.

We have added the analysis to supplement figure S3b and include it in the text on lines 111-112 and 115-117.

5) In the Introduction and (particularly) Discussion, I find the link between SSP and tree regeneration and future recovery after disturbances such as fire a bit far-fetched. Tree SSP does not have much implications for regeneration: if it did, only the tree species with high SSP would survive. Instead, the current ms nicely shows (and I think for the first time on such a scale) that tree SSP varies across several orders of magnitude, thereby emphasizing how widely different the regeneration strategies across tree species are. All these tree species survive in their own habitats, suggesting that a very wide range of different regeneration strategies are possible, depending on life history strategy, habitat type and (probably) disturbance regime. Links to the latter remain speculative however, as the current ms did not evaluate this.

We believe this is a miss-interpretation that is clarified with this revision. The link between seed size and seed number is one of the most important concepts in ecology. It plays a large role in discussions of disturbance recovery. We have added citations going back to Darwin, but we could have added many more. Reproductive success can be achieved by a range of different strategies involving seed size, numbers, fruit types, and fertility responses. The comment suggests there could be evidence that SSP has little to do with tree regeneration and recovery from disturbance; if so, it would be useful to have some citations; SSP has not previously been quantified, but a legacy of studies implicates its importance.

We hope that the revisions, including the many citations alleviate any confusion. Our Introduction highlights the shift to landscape-scale disturbances with global change where they didn't previously exist. We would be remiss to ignore how this changes the adaptive balance for many-small versus few-large seeds. We believe that the revision is fully supported by our data and the citations we include.

From a methodological perspective, I am missing important pieces of information. I find the Methods part particularly sparse of information. The section limits may not allow sufficient word space to provide all relevant details, but in that case I suggest to provide this information in Online Material. In particular, it is unclear to me:

6) Why the selected covariates have been selected. Some of them seem part of hypotheses (e.g., CEC) while most others do not. What was the reason behind including them? I can imagine some (e.g. shade class) can account for differences in SSP, but for others (e.g. slope) I find it hard to imagine such a relation. Please explain.

Because this methodology has been so widely published, we were reluctant to include too much repetition here with papers, most recently the detailed modeling and diagnostics in the Supplement to Clark et al., *Ecol Monogr* (2019). On line 232 we explain the importance of including all covariates that explain important variation, whether or not they bear directly on hypotheses. Accounting for the variation explained by other variables is common for epidemiological and ecological data.

7) How were seed crop counts and seed trap data analyzed? Were they analyzed separately, or combined? If the latter, how?

8) If the latter, how were seed trap data converted to match crop counts? Also, how were seed trap data allocated to individual trees (needed to obtain data on basal area)?

9) L234-235 "Maturation probability ρ accounts for the immature state (for small trees) and failed crop in larger trees": How? Based on what data?

We addressed 7), 8), and 9) together. This is a single model fit to all data types. We added modeling details in sections beginning on lines 218-230, 232-251, 263-286 and added figure 5. The joint modeling of seed traps and crop counts is described on lines 218-226 and 264-269. To avoid any inconvenience for reviewers, the full description is here,

<https://esajournals.onlinelibrary.wiley.com/doi/10.1002/ecm.1381>

with details here:

<https://esajournals.onlinelibrary.wiley.com/action/downloadSupplement?doi=10.1002%2Fecm.1381&file=ecm1381-sup-0001-AppendixS1.pdf>

Of course, these citations are included in the manuscript.

We added Figure 5 and a long list of citations from our work and others demonstrating that these models are effective. In particular, the Supplement link above includes diagnostics, showing recovery of parameters used to simulate data and predict the data. Figure 5 shows

how maturation status and conditional fecundity (seed production given that a tree is mature) are informed by seed traps and crop counts. We believe that the extensive revisions to this section will clarify the approach.

Finally, I have one important methodological comment:

L218-219: "Genus or family-level means were used where seed mass is missing at species level" this is indeed frequently done but my experience is that for seed size this method is highly unreliable. Seed mass differences of several orders of magnitude exist among species within the same family, and within the same genus this is even possible. By replacing species with lacking data by inappropriate estimates known to vary so widely within genera and families, data reliability is unnecessarily reduced. I strongly recommend to remove species for which seed mass is unknown at the species level from the analysis. I am expecting that for most tree species, seed mass is known as this is one of the most widely available seed traits, so that this will not greatly affect the number of tree species to be included in the analysis.

To address this concern, we repeated the analysis after removing species that use genus- and family-level seed mass. We added new figures in the supplementary materials (Fig. S8 and Fig. S9), which is discussed on lines 258-260.

Some more detailed comments:

L27: alpha is the log₁₀ (SSP at seed mass equal to 1 g), I presume?

That is correct, we believe this is clear in the revision.

Attached datafile: I only find data for 646 species, the other species contain zeroes for both crop counts and trapped seeds. Where do the data on the remaining species in the analysis (714-646= 68 species) come from?

In seed trap studies, seeds may be only assigned to genus, but the species identity is known for the trees. The MASTIF model estimates the fraction coming from each tree and thus, the species. Again, this is detailed in the full model description which is available here:

<https://esajournals.onlinelibrary.wiley.com/doi/10.1002/ecm.1381>

with details here:

<https://esajournals.onlinelibrary.wiley.com/action/downloadSupplement?doi=10.1002%2Fecm.1381&file=ecm1381-sup-0001-AppendixS1.pdf>

The file is regenerated with genus as a column in the new .csv file.

Reviewer #3 (Remarks to the Author):

This is a very interesting and complex paper. I do not completely understand all the details of your analyses, but overall I am impressed and have nothing in the way of criticism. The implications are clear: there is little to no evidence for a major tradeoff in seed production vs. seed size. The biggest criticism I have is this: now what? I recognize that part of this limitation

may be constraints imposed by the journal, which I am not very familiar with, but ...

imagine I have a student interested in this problem. Clearly, I should not advise him to pursue any kind of traditional tradeoff to explain variation in seed production, although I do note that an important factor you have not been able to consider is spatial variation: big seeds also need to be moved away from the parent. Where do you think we should go instead? I wish you had attempted this kind of analysis.

It is true that this analysis does not address spatial variation. We have included additional examples of seed size versus seed number for typical tree and seed sizes on lines 102 and 149. We believe that these examples provide a sense for the implications of the analysis that can apply to everyday experience.

Overall, very well done. Congratulations.

Eric Ribbens

Reviewers' Comments:

Reviewer #1:

Remarks to the Author:

The authors have addressed all my previous comments so I recommend the manuscript should be accepted. Francesco Chianucci

Reviewer #2:

None

Reviewer #1 (Remarks to the Author):

The authors have addressed all my previous comments so I recommend the manuscript should be accepted. Francesco Chianucci

We appreciated the efforts and comments from all reviewers, which have greatly improved the manuscript.